# Blockwise Parallel Transformers for Large Context Models

**Hao Liu**
UC Berkeley
hao.liu@cs.berkeley.edu

**Pieter Abbeel**
UC Berkeley
pabbeel@cs.berkeley.edu

## Abstract

Transformers have emerged as the cornerstone of state-of-the-art natural language processing models, showcasing exceptional performance across a wide range of AI applications. However, the memory demands posed by the self-attention mechanism and the large feedforward network in Transformers limit their ability to handle long sequences, thereby creating challenges for tasks involving multiple long sequences or long-term dependencies. We present a distinct approach, Blockwise Parallel transformers (BPT), that leverages blockwise computation of self-attention and feedforward network fusion to minimize memory costs. By processing longer input sequences while maintaining memory efficiency, BPT enables training sequences 32 times longer than vanilla Transformers and up to 4 times longer than previous memory-efficient methods. Extensive experiments on language modeling and reinforcement learning tasks demonstrate the effectiveness of BPT in reducing memory requirements and improving performance.

## 1 Introduction

Transformers [54] have become the backbone of many state-of-the-art natural language processing models [15, 45, 5, 37]. They have demonstrated impressive performance across a wide range of AI problems, including language modeling, machine translation, image captioning, and protein folding [41, 49, 32, 45, 5, 47, 9]. Transformers achieve this success through their architecture design that uses self-attention and position-wise feedforward mechanisms. These components facilitate the efficient capture of long-range dependencies between input tokens, enabling scalability in terms of context length and model size through highly parallel computations.

However, the memory requirements of Transformers limit their ability to handle long sequences, which is necessary for many AI problems, such as high-resolution images, podcasts, code, or books and especially those that involve multiple long sequences or long-term dependencies [10, 7, 41, 7, 36, 29, 49, 32, 1]. The quadratic self-attention and the large feed forward network of Transformers require a large amount of memory, which makes it challenging to scale to longer input sequences. This limitation has led to various techniques proposed to reduce the memory requirements of Transformers, including sparse-approximation, low-rank approximation, and low precision approximation [see e.g. 53, 24, 22, 11, 25, 38, 56].

One distinct line of research does not rely on approximation but instead focuses on computing exact self-attention with linear memory complexity. This approach leverages the observation that the softmax matrix in self-attention can be computed without materializing the full matrix [39]. This technique has led to the development of FlashAttention [14] and Memory Efficient Attention [44]. Both methods propose a blockwise computation of the self-attention softmax, demonstrating reduced memory requirements.

37th Conference on Neural Information Processing Systems (NeurIPS 2023).

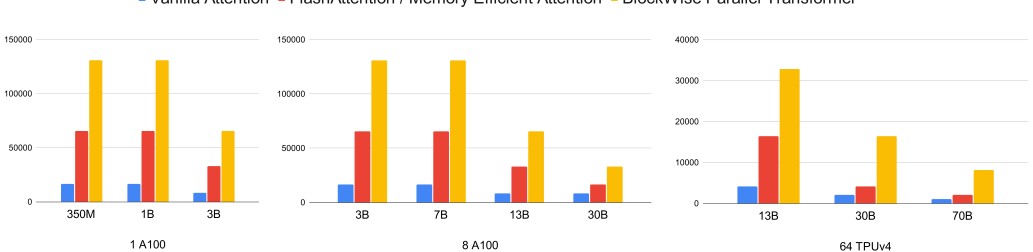

Figure 1: Maximum context length during training time with the GPT model using different methods. Model sizes range from 1B to 70B. Figures (A), (B), and (C) show evaluation using one, eight A100, and 64 TPUv4, respectively, with a single sequence. Our method enables training sequences 32 times longer than vanilla attention-based transformers [54], and 2 to 4 times longer than FlashAttention [14] and Memory Efficient Attention [44]. Section 3.1 provides a memory cost breakdown.

Despite the resulting reduced memory requirements of the self-attention block in transformers models, a significant challenge still arises from the feedforward network. This network contains a large number of parameters and produces high-dimensional intermediate vectors, resulting in substantial memory requirements. This issue is becomes the key memory challenge once employing memory-efficient attention mechanisms. Consequently, training Transformers on longer context lengths and scaling up transformers models become significantly hindered due to the overwhelming memory demands imposed by the feedforward network.

To address this challenge, we make an important observation: when self-attention is computed in a blockwise manner to reduce memory requirements, it becomes feasible to merge the computation of the feedforward network. This eliminates the need to wait for the self-attention computation to finish before performing the feedforward step on the entire sequence. By computing the feedforward network on a block-by-block basis, we effectively reduce the memory cost associated with the feedforward network. This process involves the utilization of two nested loops over the input sequence blocks. In the outer loop, we iterate over each block and compute the query. In the inner loop, we iterate over each block to calculate the key and value. These key-value pairs, along with the query, are then used to compute the blockwise attention specific to the corresponding input block. This blockwise attention is subsequently used to calculate the output of the feedforward network, followed by a residual connection. This approach enables us to process longer input sequences while maintaining lower memory budget. Since our approach performs blockwise parallel computation and fuses the feedforward and self-attention computations, we name our method the Blockwise Parallel transformers (BPT).

We evaluate the effectiveness of our approach on several benchmarks, including language modeling and reinforcement learning. Our experiments show that BPT can reduce the memory requirements of Transformers, enabling us to train 32 times longer sequence than vanilla attention [54] based GPT models and up to 4 times longer sequence than prior state-of-the-arts FlashAttention [14] and Memory Efficient Attention [44]. Furthermore, we demonstrate the application of BPT on the task of traning transformers based RL agent. By conditioning on multiple trajectories, BPT significantly improves the performance and achieves better results on challenging RL benchmarks. We believe that our approach has the potential to enable the training and evaluation of more complex models that require longer input sequences, which could lead to further breakthroughs in AI research.

Our contributions are twofold: (a) proposing a blockwise computation of self-attention and feedforward approach that enables 32 times longer and up to 4 times longer context lengths than vanilla transformers and previous memory-efficient Transformers, respectively, and (b) demonstrating the effectiveness of our approach through extensive experiments.

## 2   Memory Bottleneck of Transformer

Given input sequences $Q, K, V \in \mathbb{R}^{s \times d}$ where $s$ is the sequence length and $d$ is the head dimension. We compute the matrix of outputs as:

$$\text{Attention}(Q, K, V) = \text{softmax}(\frac{QK^T}{\sqrt{d}})V, \tag{1}$$

where softmax is applied row-wise. Standard attention implementations materialize the matrices $QK^T$ and $\text{softmax}(\frac{QK^T}{\sqrt{d}})$ to HBM, which takes $O(s^2)$ memory, so the overall space complexity is $O(s^2)$. There has been a large body of work trying to reduce memory usage of self-attention by using online softmax [39, 44, 14] to reduce memory cost of self-attention by preventing it from full materialization. And these approaches reduce memory footprint from $O(s^2)$ to $O(s)$. However, the large feedforward layers have been overlooked.

In addition to attention sub-layers, each of the attention layers is accomplished with a fully connected feedforward network, which is applied to each position separately and identically. This consists of two linear transformations with a ReLU activation in between.

$$\text{FFN}(x) = \max(0, xW_1 + b_1)W_2 + b_2 \tag{2}$$

While the linear transformations are the same across different positions, they use different parameters from layer to layer. The large size of the feedforward network requires substantial memory resources, and this becomes even more pronounced when dealing with large context sizes. See Section 3.1 for analysis of memory cost associated with transformers.

# 3 Blockwise Parallel for Large Context Models

Self-attention can be computed in a blockwise manner without materializing the softmax attention matrix $\text{softmax}(QK^T)$ [39, 14, 44]. This approach involves splitting the sequences $Q \in \mathbb{R}^{s \times d}$ into $B_q$ blocks and sequences $K, V \in \mathbb{R}^{s \times d}$ into $B_{kv}$ blocks. For each query block, the blockwise attention $\text{Attention}(Q, K, V)$ can be computed by iterating over all key-value blocks. Once the blockwise attention is computed, the global attention matrix can be obtained by scaling the blockwise attention using the difference between the blockwise and global softmax normalization constants [39]. This is achieved by keeping track of normalization statistics and combining them from all blocks to scale each block accordingly. For a specific query block $Q_i$, $1 \le i \le B_q$, the corresponding attention output can be computed by scaling each blockwise attention as follows:

$$\text{Attention}(Q_i, K, V) = \text{Scaling}(\{\exp(Q_i K_j^T)V_j\}_{j=1}^{B_{kv}}). \tag{3}$$

The scaling operation scales each blockwise attention based on the difference between the blockwise maximum and the global maximum:

$$\text{Attention}(Q_i, K_j, V_j) = \exp\big(Q_i K_j^T - \max(Q_i K_j^T)\big) / \sum \exp\big(Q_i K_j^T - \max(Q_i K_j^T)\big)$$

$$\max_i = \max\big(\max(Q_i K_1^T), \ldots, \max(Q_i K_B^T)\big)$$

$$\text{Attention}(Q_i, K, V) = \big[\exp(Q_i K_j^T - \max_i)\, \text{Attention}(Q_i, K_j, V_j)\big]_{j=1}^{B_{kv}}.$$

This blockwise self-attention computation eliminates the need to materialize the full attention matrix of size $O(n^2)$, resulting in significant memory savings.

We observe that the blockwise computation is not limited to self-attention but can also be applied to the feedforward network. For each query block, after iterating over the key and value blocks, the feedforward network can be computed along with a residual connection, completing the attention and feedforward network computation for that query block. This means that the model does not need to compute the feedforward network on the full sequence, but rather on intermediate blocks, resulting in memory savings. The computation for a query block is given by:

$$\text{Output}_i = \text{FFN}\big(\text{Attention}(Q_i, K, V) + Q_i\big) + \text{Attention}(Q_i, K, V) + Q_i.$$

Therefore, the output for each block consists of the feedforward network, self-attention, and residual connection computed in a blockwise manner.

It is worth mentioning that for large models, the memory cost of the feedforward network on the full sequence can be much larger than the memory efficient attention. Therefore computing the feedforward network on the same block as attention can significantly reduce memory cost, and it also reduces data movements, contributing to overall computational efficiency. Moreover, we should remark that blockwise parallelism can be directly applied to the final cross entropy loss, which can further minimize memory cost. The full process of our framework, coined as BPT, is summarized in Algorithm 1.

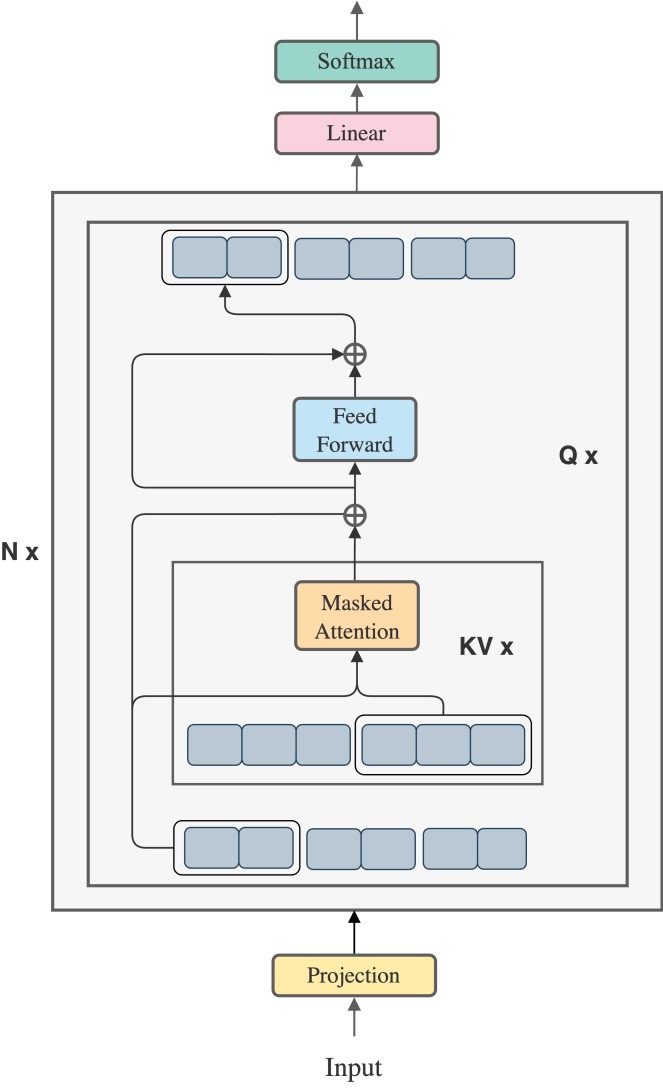

Figure 2: We use the same model architecture as the original transformers but with a different way of organizing the compute. In the diagram, we explain this by showing that for the bottom first incoming input block, we project it into query; then we iterate over the same input sequence positioned above the bottom row, and project it to key and value. These query, key and value are used to compute self-attention (yellow box), whose output is pass to feedforward network (cyan box), followed by a residual connection. In our proposed approach, this process is then repeated for the other incoming input blocks.

## 3.1 Analysis of Memory Cost

We present an analysis of memory cost across different transformers architectures: the Vanilla Transformer, the memory-efficient / Flash Attention variant, and BPT.

**Vanilla Transformers:**

Attention: For $Q, K, V$, saving their input $x$ needs $2bsh$ bytes, where $b$ is batch size, $s$ is sequence length, and $h$ is hidden dimension. For $QK^T$ matmul, saving activations $Q$ and $K$ needs $4bsh$ bytes. For softmax$(QK^T)$, saving input $QK^T$ needs $2bs^2a$ bytes, where $a$ is the number of attention heads. For mask and dropout, saving mask needs $bs^2a$ bytes. For score $\times V$, saving score needs $2bs^2a$ bytes, and saving $V$ needs $2bsh$ bytes. For output projection and dropout, saving the input needs $2bsh$ bytes, and saving dropout mask needs $bsh$ bytes. The maximum attention activation size of attention is $O(s^2)$ with checkpointing.

---

**Algorithm 1** Reduce memory cost with BPT.

---

**Required:** Input sequence $x$. Number of query blocks $B_q$. Number of key and value blocks $B_{kv}$.
Initialize
Project input sequence $x$ into query, key and value.
Split query sequence into $B_q$ of query input blocks.
Split key and value sequences into $B_{kv}$ of key-value input blocks.
**for** $outer = 1$ **to** $B_q$ **do**
    Choose the $outer$-th query.
    **for** $inner = 1$ **to** $B_{kv}$ **do**
        Choose the $inner$-th key and $inner$-th value block.
        Compute attention using query, key and value, and record normalization statistics.
    **end for**
    Combine each blocks by scaling them to get attention output for the $outer$-th input block.
    Compute feedforward on attention output and add residual connection.
**end for**

---

FFN: For the first linear layer, saving input needs $2bsh$ bytes. For activation, saving input needs $8bsh$ bytes. For the second linear layer, saving input needs $8bsh$ bytes. For dropout, saving the mask needs $bsh$ bytes. With checkpointing, the maximum activation size of FFN is $8bsh$ bytes.

Consequently, for a large context length, the memory cost of activation in vanilla transformers is $O(s^2)$.

**BPT:**

Attention: Since BPT does not materialize full attention and instead computes it blockwise, it needs to store intermediate blockwise activations in the key-value loop, which has a maximum activation size of $4bch$ with checkpointing. Additionally, it needs to store $q$ output activations for the query loop, which requires $2bsh$ bytes. Since $s \gg c$, the maximum activation size is $2bsh$.

FFN: When iterating the FFN over blocks, BPT needs to save the following activations: For the first linear layer, saving input needs $2bch$ bytes. For activation, saving input needs $8bch$ bytes. For the second linear layer, saving input needs $8bch$ bytes. For dropout, saving the mask needs $bch$ bytes. In total, $19bch$ bytes are needed. Additionally, storing the output of the for loop requires $2bsh$ bytes. Therefore, the maximum FFN activation size is $2bsh$.

Consequently, each BPT layer's memory cost of activation is $2bsh$.

**Memory-Efficient / Flash Attention:**

Attention: Similar to BPT attention, the maximum activation size is $2bsh$.

FFN: Similar to the vanilla FFN, the maximum activation size is $8bsh$.

Consequently, each Flash Attention layer's memory cost is $8bsh$.

Comparing the activation memory of Flash Attention/Memory-Efficient transformers with BPT, we see that BPT offers $8bsh/2bsh = 4$ times memory saving. By taking into account other factors of memory cost such as model parameters and optimizer states, BPT allows training with context lengths 2-4 times larger than prior state-of-the-arts.

### 3.2 Why Blockwise Parallel

The utilization of blockwise parallelization may raise questions about the effectiveness of running parallel computers, as computation can become sequential between blocks. However, the benefits of blockwise parallelization depend on the model size and hardware configuration. In cases where the model is large or the context length is extremely long, a block may reach its maximum arithmetic density, making it impractical to execute the original full-length sequence in parallel. In such scenarios, blockwise parallelization treats the long sequence as short ones, allowing dealing with large models and effectively enabling large context size. Moreover, using blockwise parallelization allows us to avoid waiting for the completion of self-attention and allocating a significant amount of memory solely for feed-forward network computation.

Another notable advantage of blockwise parallelization is its ability to leverage hardware with significantly faster SRAM speed compared to HBM speed. For instance, in Nvidia GPUs, SRAM is an order of magnitude faster than HBM, while in Google TPUs, SRAM also offers higher speed than HBM. By utilizing blockwise parallelization, we can tap into the increased speed of SRAM, thereby reducing communication costs and increasing throughput. This advantage aligns with memory efficient self-attention approaches [14, 44].

### 3.3 Implementation

Algorithm 1 provides the pseudocode of the algorithm. Figure 3 in Appendix shows a Jax implementation optimized for simplicity. The full code of BPT is provided at GitHub [1] which supports large-scale distributed training of large context models using BPT.

The `blockwise_ffn` function begins by accepting a rematerialized feed forward module, inputs and chunk size. The `remat_ffn` compute feedforward on inputs with checkpointing, *i.e.*without saving intermediates. The `scan_ffn` function is then used to scan over input sequences and generate outputs.

The `blockwise_attn` function process query, key, and value to produce attention blockwise. The `scan_attention` function is defined, which computes the attention weights between the query vector and key-value pairs from another chunk. This is done by applying the `scan_kv_block` function to the key-value chunk, calculating the dot product between the query and key vectors, and then adding a bias term. The bias term introduces a positional bias between different chunks based on their indices without materializing the full matrix. The softmax function is then applied to the attention weights in a numerically stable manner, using the max-score trick to avoid large exponentiation results.

Finally, BPT combines the outputs from all chunks, normalizes them using their max-score-adjusted weights, and passes them through a feed-forward neural network (`blockwise_ffn`). The final output is the sum of the feed-forward output, the attention output, and the original input.

## 4 Setting

We evaluate the impact of using BPT in improving large transformers models by benchmarking memory requirement, maximum sequence length and throughout speed. We show apply BPT to reinforcement learning as an application.

**Model Configuration.** Our study is built upon the GPT architecture. Table 1 provides a overview of the model sizes considered in our experiments.

**Baselines.** We evaluate our method by comparing it with vanilla Transformer [54] which is denoted as "Vanilla", and FlashAttention [14] and Memory Efficient Attention [44] which are state-of-the-art memory efficient attention, we denote them as "MemoryEfficient" in our experiments. All methods use the same gradient checkpointing in the experiments.

**Datasets.** We consider two datasets for evaluation purposes. Including pretraining on OpenWebText dataset and large context reinforcement learning on ExoRL.

- **OpenWebText.** The OpenWebText dataset [18] is a large and diverse collection of web pages that has been filtered and cleaned for use in natural language processing (NLP) tasks. The dataset consists of over 6 billion tokens from more than 40 million web pages, covering a wide range of topics and genres.

- **ExoRL.** The ExoRL [58] dataset is based on unlabeled exploratory data collected by running unsupervised RL algorithms. For each environment, it comes with eight different unsupervised data collection algorithms, taken from from URLB [28]. The datasets are collected by unsupervised RL and then relabeled using task reward function. The resulting mixed dataset consists of 8 millions timesteps (8000 episodes), with each episode spanning a length of 1000 steps.

**Training Configuration.** Our main baselines are vanilla attention [54], which computes self-attention by materializing the attention matrix and computes the feedforward network normally. We also

---

[1] `https://github.com/lhao499/llm_large_context`

Table 1: Sizes and architectures of the models which we evaluated in experiments.

| Model Name | $n_{\text{params}}$ | $n_{\text{layers}}$ | $d_{\text{model}}$ | $n_{\text{heads}}$ | $d_{\text{head}}$ |
|---|---|---|---|---|---|
| GPT 1B | 1.3B | 24 | 2048 | 16 | 128 |
| GPT 3B | 2.7B | 32 | 2560 | 32 | 80 |
| GPT 7B | 6.7B | 32 | 4096 | 32 | 128 |
| GPT 13B | 13.0B | 40 | 5140 | 40 | 128 |
| GPT 30B | 30.0B | 48 | 7168 | 56 | 128 |
| GPT 70B | 70.0B | 80 | 8192 | 64 | 128 |

consider two prior state-of-the-art memory-efficient methods, namely FlashAttention [14], which focuses on GPU efficiency, and Memory Efficient Attention [44], which focuses on TPU efficiency. Since they share a similar idea, for notation simplicity, we refer to them as FlashAttention in our experiments. We tune the block size for both the baselines and BPT, and report the best results achieved by each. The experiments are on NVIDIA 80GB A100 GPUs, we consider both single GPU for smaller model training and 8 GPUs settings for model parallel training. We also experiment with scaling up model on 64 TPUv4.

We note that no data parallelism is considered in our evaluations since our approach is independent of data parallelism. As a result, the batch sizes used in our analysis are much lower than the ones used for the end-to-end training. All of our results are obtained using full precision instead of mixed precision.

## 5   Results

In our experiments, our primary objective is to comprehensively evaluate the performance of BPT across multiple key metrics, including maximum sequence length, memory usage, and throughput. Moreover, we extend the applicability of BPT to reinforcement learning and evaluate its effectiveness in large context application.

Table 2: Maximum context length during training with different methods. BPT enables training 2-4 times longer sequence length than FlashAttention / Memory Efficient Attention, and up to 32 times longer sequence length than vanilla attention.

| 1 A100 | PartitionSpec | Vanilla Attention | MemoryEfficient | Blockwise Parallel |
|---|---|---|---|---|
| 350M | (1,1,1) | 16K (16384) | 65K (65536) | 131K (131072) |
| 1B | (1,1,1) | 16K (16384) | 65K (65536) | 131K (131072) |
| 3B | (1,1,1) | 8K (8192) | 16K (16384) | 65K (65536) |

| 8 A100 | PartitionSpec | Vanilla Attention | MemoryEfficient | Blockwise Parallel |
|---|---|---|---|---|
| 3B | (1,1,8) | 16K (16384) | 65K (65536) | 131K (131072) |
| 7B | (1,1,8) | 16K (16384) | 65K (65536) | 131K (131072) |
| 13B | (1,1,8) | 8K (8192) | 33K (32768) | 65K (65536) |
| 30B | (1,1,8) | 8K (8192) | 16K (16384) | 65K (65536) |

| 64 TPUv4 | PartitionSpec | Vanilla Attention | MemoryEfficient | Blockwise Parallel |
|---|---|---|---|---|
| 13B | (1,1,64) | 4K (4096) | 16K (16384) | 33K (32768) |
| 30B | (1,1,64) | 2K (2048) | 4K (4096) | 16K (16384) |
| 70B | (1,1,64) | 1k (1024) | 2K (2048) | 8K (8192) |

### 5.1   Evaluation of Context Length

We present experimental results comparing the maximum training sequence lengths achieved using three different attention mechanisms: Vanilla, MemoryEfficient, and Blockwise Parallel. Table 2 summarizes the findings. On one A100 GPU, Vanilla transformers supports a maximum training sequence length of 16K for 1B parameters and 8K for 3B parameters. In contrast, MemoryEfficient enables longer sequences of 65K for 1B parameters and 16K for 3B parameters. Notably, our proposed method, Blockwise Parallel, surpasses both methods, achieving a maximum sequence

length of 131K for 1B parameters and 3B parameters. Moving on larger models, Blockwise Parallel again outperforms the other two methods, allowing training sequences of 65K for 30B large model on 8 GPUs and 8K for 70B large model on 64 TPUv4, which are two and four times longer than MemoryEfficient, respectively.

Table 3 shows the analysis of memory usage across different settings with three distinct approaches: Vanilla Transformer, MemoryEfficient, and our proposed method, BPT. It is evident that Vanilla transformers consumes the highest amount of memory, while MemoryEfficient and BPT offer notable improvements in memory optimization. Notably, our BPT technique consistently outperforms both Vanilla transformers and MemoryEfficient in all settings, showcasing memory efficiency.

Table 3: Memory usage comparison for different settings. "oom" denotes out of memory.

| Setting | 3B on A100 | | | 13B on 8 A100 | | |
|---|---|---|---|---|---|---|
| Context Length | Vanilla | MemoryEfficient | BPT | Vanilla | MemoryEfficient | BPT |
| 8192 | 64GB | 44GB | 43GB | 59GB | 44GB | 42GB |
| 16384 | oom | 47GB | 45GB | oom | 46GB | 45GB |
| 32768 | oom | 55GB | 52GB | oom | 55GB | 52GB |
| 65536 | oom | 75GB | 70GB | oom | 75GB | 68GB |
| 131072 | oom | oom | 79GB | oom | oom | 78GB |

## 5.2  Evaluation on Throughput and Speed

In Table 4, we present a comparison of the throughput achieved by different attention mechanisms on the GPT-XL (1B) model trained on the OpenWebText dataset using 8 GPUs. Throughput is measured as number of tokens processed per device per second. We evaluate the performance at various context lengths, including 2K, 8K, 16K, 33K, and 65K tokens. Our proposed method achieves competitive throughput as MemeoryEfficient mechanism, and surpasses the Vanilla transformer, achieving 1.17x speedup at context length 8k and 1.2x speedup at context length 16k. At context length 32K and 64K, our method maintains high throughput and training speed, while the alternatives cannot train due to running out of memory. This demonstrates the scalability and efficiency of our proposed method, allowing it to effectively handle large context lengths without compromising on throughput and training speed.

## 5.3  Evaluation on Reinforcement Learning

In this section, we present the results of applying BPT to improve the performance of transformers in reinforcement learning (RL). We report our results in Table 5, where we evaluate our proposed model on the ExoRL benchmark across six different tasks. On ExoRL, we report the cumulative return, as per ExoRL [58]. The numbers of BC, DT [6] and AT [34] are from the ExoRL and AT paper. AT + ME and AT + BPT numbers are run by ourselves. Since the ExoRL data is significantly more diverse than D4RL because it is collected using various unsupervised RL algorithms [28, 35, 33], it is found that TD learning performs best while behavior cloning struggles [58]. AT [34] shows that conditioning transformers on multiple trajectories with relabeled target return can significantly outperforms behavior cloning approaches BC-10% and DT, and achieves competitive results with TD learning. For more details, please refer to their papers. We are interested in applying BPT to improve the performance of AT by conditioning on a 32 trajectories rather than 4 trajectories in original work. It is worth noting that each trajectory has $1000 \times 4$ length where 1000 is sequence length while 4 is return-state-action-reward, making training 32 trajectories with 350M size model infeasible for both Vanilla and MemoryEfficient. Results in Table 5 show that, by scaling the sequence length, AT + BPT consistently outperforms the original transformers model in all six tasks, achieving a total average return of 111.13 compared to the original transformers model's total average return of 83.02

## 6  Related Work

Transformers have garnered significant attention in the field of natural language processing (NLP) and have become the basis for numerous state-of-the-art models. Several works have explored memory-

Table 4: Throughput comparison on GPT-XL (1B) using OpenWebText dataset. Throughput is measured as tokens processed per second. 'oom' denotes running out of memory, 'na' denotes results not available because we early terminated these runs to reduce compute cost.

| Model | Context Len | Val Loss | Throughput | Speed up |
|---|---|---|---|---|
| Vanila transformers | 2048 | 2.46 | 3827 | 1x |
| MemoryEfficient | 2048 | 2.46 | 4371 | 1.14x |
| **Blockwise Parallel** | 2048 | 2.46 | 3985 | 1.04x |
| Vanila transformers | 4096 | 2.44 | 2340 | 1x |
| MemoryEfficient | 4096 | 2.44 | 2567 | 1.1x |
| **Blockwise Parallel** | 4096 | 2.44 | 2687 | 1.15x |
| Vanila transformers | 8192 | 2.43 | 2455 | 1x |
| MemoryEfficient | 8192 | 2.43 | 2781 | 1.13x |
| **Blockwise Parallel** | 8192 | 2.43 | 2875 | 1.17x |
| Vanila transformers | 16384 | 2.41 | 1701 | 1x |
| MemoryEfficient | 16384 | 2.41 | 1889 | 1.11x |
| **Blockwise Parallel** | 16384 | 2.41 | 2045 | 1.2x |
| Vanila transformers | 32768 | oom | oom | oom |
| MemoryEfficient | 32768 | na | 810 | 1x |
| **Blockwise Parallel** | 32768 | na | 857 | 1.1x |
| Vanila transformers | 65536 | oom | oom | oom |
| MemoryEfficient | 65536 | oom | oom | oom |
| **Blockwise Parallel** | 65536 | na | 600 | 1x |

Table 5: Application of BPT on improving transformers in RL. All the baselines use vanilla attention. AT + ME denotes using "MemoryEfficient". AT + BPT denotes using Blockwise Parallel.

| ExoRL benchmark | BC-10% | DT | AT | AT | AT + ME | AT + BPT |
|---|---|---|---|---|---|---|
| **Task** | | | N Trajs = 4 | N Trajs = 32 | N Trajs = 32 | N Trajs = 32 |
| Walker Stand | 52.91 | 34.54 | 68.55 | oom | oom | 95.45 |
| Walker Run | 34.81 | 49.82 | 88.56 | oom | oom | 105.88 |
| Walker Walk | 13.53 | 34.94 | 64.56 | oom | oom | 78.56 |
| Cheetah Run | 34.66 | 67.53 | 125.68 | oom | oom | 178.75 |
| Jaco Reach | 23.95 | 18.64 | 52.98 | oom | oom | 87.56 |
| Cartpole Swingup | 56.82 | 67.56 | 97.81 | oom | oom | 120.56 |
| **Total Average** | 36.11 | 45.51 | 83.02 | oom | oom | 111.13 |

efficient techniques to address the memory limitations of Transformers and enable their application to longer input sequences. One line of research focuses on various approximation techniques or compressing along the sequence dimension [see e.g. 24, 12, 14, 4, 44, 56, 38, 25]. Other works explored replacing attention [19, 20, 43, 23, 3, 59, 42, 55]. Another line of work explores partitioning the large hidden dimension of the feedforward network into parts and retrieving only one part per token [30, 50, 17, 26, 60, 62]. Additionally, extending the context by attending over states from previous sequences has been explored [13, 46], as well as combining local and global contexts [21, 11]. For a comprehensive review of these techniques, we recommend referring to the surveys by Tay et al. [53], Narang et al. [40], Tay et al. [52]. Several studies explored sharding large model on distributed devices tensor, data, or sequence parallelism [51, 16, 57, 27, 61, 31, 48]. Ours shares similarities with the sequence parallelism [27] where sequences are distributed across devices, in contrast, ours implements blockwise computation on sequences for each device. This creates an orthogonal relationship between our method and sequence parallelism, allowing for straightforward combination. In addition, our methodology is compatible with both tensor and data parallelism. Another direction involves computing exact self-attention in a blockwise manner using the tiling technique [39]. This approach has led to the development of memory efficient attention mechanisms [14, 44]. In line with these advancements, our work falls into this category. We propose computing both the feedforward

network and self-attention in a blockwise manner, resulting in a significant reduction in memory requirements.

# 7 Conclusion

In conclusion, we propose a blockwise parallelization approach to reduce the memory requirements of Transformers, the backbone of state-of-the-art AI models. Our approach enables processing longer input sequences while maintaining or improving performance. Through extensive experiments, we demonstrate its effectiveness, achieving up to 4x memory reduction than memory-efficient Transformers. Our contributions include a practical method for large context sizes in large transformers models. With the increasing capability of hardware, larger models and longer context length are widely used in AI research. At the same time, as we are pushing up against physics and fabrication limits, it is more important to design scaling approaches as efficient as possible to scale up large models and large context size. Our approach holds promise for training and evaluating complex models with longer input sequences, potentially driving new breakthroughs in machine learning research.

**Limitations and Future Work.** Although our method achieves state-of-the-art low memory usage for transformers models, it does have some limitations that need to be addressed:

- *Optimal performance.* While our implementation prioritizes simplicity with high-level Jax operations, optimizing low-level operations is crucial for achieving optimal performance. In future work, we suggest considering porting our method to CUDA and OpenAI Triton to achieve minimal memory cost and maximum speedup.

# Acknowledgements

This project is supported in part by ONR under N00014-21-1-2769. We would like to express our sincere appreciation to the members of the RLL Lab and Berkeley AI Lab, as well as Anselm Levskaya, Markus Rabe, Federico Lebron, Sharad Vikram, and Tri Dao for their valuable insights and contributions to this paper. We thank Google TPU Research Cloud for granting us access to TPUs.

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

# A   Experiment Details

## A.1   Evaluation of Memory

In the experimental results presented in Section 5.1, we used model parallelism to partition the model across 8 GPUs or 64 TPUv4 units. Our evaluation focused on determining the maximum achievable sequence length, using a sequence number of one. For TPUs, we utilized its default training configuration, which involved performing matmul operations in `bfloat16` format with weight accumulation in `float32`. On the other hand, for GPUs, we adopted the default setup, where all operations were performed in `float32`.

To profile memory usage, we utilized `jax.profile` and repeated the evaluation 100 times, reporting the average results. We conducted a grid search for the optimal query block size and key-value block size, considering values from the set $[16, 64, 128, 512, 1024, 2048, 4096]$. For each method, we reported the lowest memory achieved.

## A.2   Evaluation of Throughput

In the evaluation presented in Section 5.2, we split OpenWebText following the methodology of [2]. Throughput is measured as tokens per device per second. To ensure a fair comparison, we performed a grid search for the optimal query block size and key-value block size, considering values from the set $[16, 64, 128, 512, 1024, 2048, 4096]$. For gradient checkpointing [8], we additionally grid search among three commonly used checkpointing policies including `noth-ing_saveable`, `dots_saveable`, and `dots_with_no_batch_dims_saveable` for attention and use `nothing_saveable` for feedforward network (FFN). For more details, please refer to Jax documentation. We selected the best performing configuration for both baselines and our method.

The training was conducted using FSDP [16] and gradient accumulation. We used weight decay of 0.1 and utilized cosine learning rate decay with a maximum learning rate of $2.0 \times e^{-4}$. For sequence lengths of $2048, 4096, 8192, 16384$, the batch sizes in trajectories were set as $8, 4, 2, 1, 1$ respectively. We use gradient accumulation to accumulate batch size in tokens to 1 million per batch.

## A.3   Evaluation on RL

Table 6: Hyperparameters used in RL evaluation.

| Hyperparameter | Value |
|---|---|
| Number of layers | 3 |
| Number of attention heads | 1 |
| Embedding dimension | 128 |
| Activation function | ReLU |
| Batch size | 64 |
| Dropout | 0.1 |
| Learning rate | $10^{-4}$ |
| Learning rate decay | Linear warmup for $10^5$ steps |
| Grad norm clip | 0.25 |
| Weight decay | $10^{-4}$ |
| Initial desired target return at test time | 850 Walker Stand |
| | 400 Walker Run |
| | 900 Walker Walk |
| | 350 Cheetah Run |
| | 300 Jaco Reach |
| | 800 Cartpole Swingup |
| Number of trajectories during training | $4 \rightarrow 32$ |
| Number of trajectories at test time | $4 \rightarrow 16$ |

In the experiment presented in Section 5.3, we followed the prior work's setting for learning rate, batch size, and other hyperparameters, while modifying the number of trajectories. The specific hyperparameters are provided in Table 6. The original agentic transformers used 4 trajectories during training, we increase the number to 32.

During testing, increasing the number of trajectories has been shown to improve performance. However, performing autoregressive sampling over a large number of trajectories (e.g., $64 \times 1000 \times 4$

total number of tokens) can be computationally slow. To reduce the sampling time, we limited the rollout to 16 trajectories.

```python
1   def blockwise_ffn(remat_ffn, inputs, chunk_size, deterministic):
2       # remat_ffn: a rematerialized ffn
3       inputs = rearrange(inputs, 'b (c n) d -> b c n d', c=chunk_size)
4       def scan_ffn(remat_ffn, carry, hidden_states):
5           outputs = remat_ffn(hidden_states, deterministic=deterministic)
6           return carry, outputs
7       scan_axis = inputs.ndim - 2
8       _, res = nn.scan(
9           scan_ffn,
10          variable_broadcast="params",
11          split_rngs={"params": False, "dropout": True},
12          in_axes=scan_axis,
13          out_axes=scan_axis,
14      )(remat_ffn, None, inputs)
15      res = rearrange(res, 'b c n d -> b (c n) d')
16      return res
17
18  def blockwise_attn(query, key, value, query_chunk_size,
19          key_chunk_size, dtype, policy, precision, prevent_cse):
20      query = query / jnp.sqrt(query.shape[-1]).astype(dtype)
21      query = rearrange(query, 'b (c n) h d -> n b c h d', c=query_chunk_size)
22      key, value = map(lambda t: rearrange(t, 'b (c n) h d -> n b c h d',
23      c=key_chunk_size), (key, value))
24      num_q, batch, _, num_heads, dim_per_head = query.shape
25      num_kv = key.shape[0]
26      def scan_attention(args):
27          query_chunk, query_chunk_idx = args
28          @functools.partial(jax.checkpoint, prevent_cse=prevent_cse, policy=policy)
29          def scan_kv_block(carry, args):
30              key_chunk, value_chunk, key_chunk_idx = args
31              (numerator, denominator, prev_max_score) = carry
32              attn_weights = jnp.einsum('bqhd,bkhd->bqhk', query_chunk,
33              key_chunk, precision=precision)
34              bias_chunk = _chunk_bias_fn(query_chunk_idx, key_chunk_idx)
35              bias_chunk = jnp.moveaxis(bias_chunk, 1, 2)
36              attn_weights = attn_weights + bias_chunk
37
38              max_score = jnp.max(attn_weights, axis=-1, keepdims=True)
39              max_score = jnp.maximum(prev_max_score, max_score)
40              max_score = jax.lax.stop_gradient(max_score)
41              exp_weights = jnp.exp(attn_weights - max_score)
42              exp_values = jnp.einsum(
43                  'bqhv,bvhf->bqhf', exp_weights, value_chunk, precision=precision
44              )
45              correction = jnp.exp(prev_max_score - max_score)
46              numerator = numerator * correction + exp_values
47              denominator = denominator * correction + exp_weights.sum(axis=-1, keepdims=True)
48              return Carry(numerator, denominator, max_score), None
49          init_carry = Carry(
50              jnp.zeros((batch, query_chunk_size, num_heads, dim_per_head), dtype=query.dtype),
51              jnp.zeros((batch, query_chunk_size, num_heads, dim_per_head), dtype=query.dtype),
52              (-jnp.inf) * jnp.ones((batch, query_chunk_size, num_heads, 1), dtype=query.dtype),
53          )
54          (numerator, denominator, max_score), _ = lax.scan(
55              scan_kv_block, init_carry, xs=(key, value, jnp.arange(0, num_kv))
56          )
57          outputs = (numerator / denominator).astype(dtype)
58          return outputs
59      _, res = lax.scan(
60          lambda _, x: ((), scan_attention(x)),
61          (), xs=(query, jnp.arange(0, num_q))
62      )
63      res = rearrange(res, 'n b c h d -> b (n c) h d')
64      return res
```

Figure 3: Key parts of the implementation of Blockwise Parallel in Jax. The full code is available at https://github.com/lhao499/llm_large_context

