# OpenReview forum: "Blockwise Parallel Transformers for Large Context Models"
_NeurIPS.cc/2023/Conference — NeurIPS 2023 spotlight_

### Official Review · Reviewer_KeWq · 2023-07-07

**Soundness:** 3 good
**Presentation:** 4 excellent
**Contribution:** 2 fair
**Rating:** 7
**Confidence:** 5

**Summary:**

The paper extends the work in Flash Attention to apply block wise attention to not just the self attention block, but also to the feedback forward part of the transformer block. While doing so, the output of the self attention block is not stored for backprop, and recalculated. This leads to very efficient gain in memory requirements for training on large sequences.

As already shown by FlashAttention work, performing these operations at block level can take advantage of limited SRAM by scheduling operations in the right order to reduce the amount of communication thereby not just leading to memory efficiency, but also improved throughput and latency.

Experiments on multiple datasets show that this improves upon the foundation laid by FlashAttention by being able to support longer sequences along with throughput improvements.

**Strengths:**

- Efficiency of Transformer Architectures is a pretty important field, and any improvement in this area has a wide impact on various applications of Transformers.
- Memory Efficient architectures are key to the future of both research and deployment.
- The paper does a reasonable number of experiments to prove that even though the operations are done blockwise, the throughput improves because of better SRAM utilization with reduced communication.
- Table 4: By increasing the sequence length, they are also able to get better validation loss.

**Weaknesses:**

- Line 23: "such as" is repeated.
- The contribution on top of Flash Attention is pretty limited, but still the impact is pretty significant.

**Questions:**

- In the RL task, the AT baseline is for NTrajs=4, and compared with N Trajs =64 which OOMs for AT. Is it possible to evaluate on N Trajs = something in between 4 and 64 that allows AT to run without OOM but perform better than N Trajs =4 for a better comparison?
- Why is stop_gradient needed in the implementation for `max_score`?


**Limitations:**

Limitions are well noted in the paper.

---

> ### Author Rebuttal · Authors · 2023-08-09
>
> We thank the reviewer for their feedback and for a positive assessment of the work. We found the reviewer’s questions and suggestions insightful, and we list our plans to incorporate them below. Please let us know if our answers address your questions.
>
>
> > Q: In the RL task, the AT baseline is for NTrajs=4, and compared with N Trajs =64 which OOMs for AT. Is it possible to evaluate on N Trajs = something in between 4 and 64 that allows AT to run without OOM but perform better than N Trajs =4 for a better comparison?
>
> A: AT with vanilla transformer architecture runs out of memory for more than 16 trajectories. We have conducted experiments for AT with 16 trajectories. On ExoRL, increasing to 16 trajectories improves results significantly compared to 4 trajectories. AT + BPT trained with 64 trajectories performs the best.
>
> | ExoRL benchmark | BC-10% | DT | AT | AT | AT | AT + FA | AT + BPT |
> |-----------------|--------|----|----|----|----|---------|---------|
> | Task            |        |    | N Trajs = 4 | N Trajs = 16 | N Trajs = 64 | N Trajs = 64 | N Trajs = 64 |
> | Walker Stand    | 52.91  | 34.54  | 148.55   | 161.45   | oom   | oom   | 195.45     |
> | Walker Run      | 34.81  | 49.82  | 98.56    | 125.44   | oom   | oom   | 145.98     |
> | Walker Walk     | 13.53  | 34.94  | 84.56    | 89.66    | oom   | oom   | 103.88     |
> | Cheetah Run     | 34.66  | 67.53  | 145.68   | 164.31   | oom   | oom   | 178.75     |
> | Jaco Reach      | 23.95  | 18.64  | 58.98    | 60.95    | oom   | oom   | 87.56      |
> | Cartpole Swingup| 56.82  | 67.56  | 187.56   | 191.34   | oom   | oom   | 220.56     |
> | Total Average   | 36.11  | 45.51  | 120.65   | 132.19   | oom   | oom   | 155.36     |
>
>
> > Q: Why is stop_gradient needed in the implementation for max_score?
>
> A: Jax compiler and XLA use heuristics to optimize computation graphs created by users code, but such optimization is oftentimes inefficient due to combinatorial possibilities. Without using the stop_gradient, we found that the compilation process becomes time-consuming. Given that the gradient is unnecessary for max_score, utilizing stop_gradient serves as a clear directive to the compiler to disregard its gradient during optimization.

---

### Official Review · Reviewer_t1yD · 2023-07-11

**Soundness:** 3 good
**Presentation:** 3 good
**Contribution:** 2 fair
**Rating:** 6
**Confidence:** 4

**Summary:**

This paper proposed to merge the computation of the FFN layers in Transformer into the block-wise computation of self-attention. This eliminates the need to wait for the self-attention computation to finish before performing the feedforward step on the entire sequence, because FFN is entirely position-wise and hence block-parallelizable.

Experiments on large language pretraining and reinforcement learning demonstrate that BPT reduces the peak memory costs and allows training large-scale Transformer models on longer sequences.

**Strengths:**

The proposed method is straight forward and well-motivated. The paper is writing well, easy to follow. Experiments show the benefits of BPT on long sequence training, especially the cost of memory.

**Weaknesses:**

The novelty of the proposed method is not that significant, and there are still some parts unclear (see questions).

**Questions:**

If I understand correctly, even the FFNs are computed in a block-wise manner, we still need to keep all the output vectors of each FFN layer (and self-attention layer) to do back-propagation. So the memory cost still increases linearly w.r.t the length of inputs. By using the activation checkpointing, we can reduce the memory cost of the hidden vectors inside FFNs (4 x model dim), but not the output of each layer. Then, why BPT is able to train 2-4 times longer sequences than FlashAttention? Just because of the memory reduction of the FFN hidden vectors?

**Limitations:**

NA.

---

> ### Author Rebuttal · Authors · 2023-08-09
>
> Thank you for the review. We found the reviewer’s questions and suggestions insightful, and we list our plans to incorporate them below. You raised a good question about how exactly does BPT help save memory. Please let us know if our answers address your questions.
>
>
> > Q: If I understand correctly, even the FFNs are computed in a block-wise manner, we still need to keep all the output vectors of each FFN layer (and self-attention layer) to do back-propagation. So the memory cost still increases linearly w.r.t the length of inputs. By using the activation checkpointing, we can reduce the memory cost of the hidden vectors inside FFNs (4 x model dim), but not the output of each layer. Then, why BPT is able to train 2-4 times longer sequences than FlashAttention? Just because of the memory reduction of the FFN hidden vectors?
>
>
> A: We provide an analysis of the memory costs associated with vanilla Transformer, the memory-efficient/flash attention variant, and BPT. This breakdown will provide you with a clear understanding of the memory usage of each architecture.
>
> **Vanilla Transformer**
>
> *Attention:*
> For Q, K, V, saving their input x needs 2bsh bytes, where b is batch size, s is sequence length, and h is hidden dimension
> For QK^T matmul, saving activation Q and K needs 4bsh bytes
> For softmax(QK^T), saving input QK^T needs 2bs ** 2a, a is number of attention heads
> For mask and dropout, saving mask needs bs ** 2a
> For score * V, saving score needs 2bs ** 2a and saving V needs 2bsh
> For output projection and dropout, saving the input needs 2bsh, saving dropout mask needs bsh
> In total attention activation size 11bsh + 5bs ** 2a
> Thus the peak attention activation size of attention is O(s ** 2) with checkpointing.
>
> *FFN:*
> For first linear layer, saving input needs 2bsh
> For activation, saving input needs 8bsh
> For second linear layer, saving input needs 8bsh
> For dropout, saving mask needs bsh
> In total MLP activation size is 19bsh
> With checkpointing, the peak activation size of FFN is 8bsh.
>
> Consequently, for large context length, the memory cost of activation in vanilla Transformer is O(s**2).
>
> **BPT**
>
> *Attention:*
> Since BPT doesn’t materialize full attention and instead computes it blockwise, it needs to store intermediate blockwise activations in the key-value for loop which has 4bch peak activation size with checkpointing, and it also needs to store q output activations for query for loop which requires 2bsh. Since s >> c, the peak activation size is 2bsh.
>
> *FFN:*
> Iterating ffn over blocks, BPT needs to save the following activations:
> For first linear layer, saving input needs 2bch
> For activation, saving input needs 8bch
> For second linear layer, saving input needs 8bch
> For dropout, saving mask needs bch
> In total 19bch, in addition, storing the output of the for loop needs 2bsh. Therefore, the peak FFN activation size is 2bsh.
>
> Consequently, each BPT layer’s memory cost of activation is 2bsh.
>
> **Memory efficient  / flash attention transformer**
>
> *Attention:*
> Same as BPT attention, the memory cost is 2bsh + 4bch.
>
> *FFN:*
> Same as vanilla FFN, the peak activation size is 8bsh.
>
> Consequently, each flashattention layer’s memory cost is 8bsh
>
> ---
> Comparing activation memory of FlashAttention/MemEff with BPT, we see that BPT offers 8bsh/2bsh = 4 times memory saving. By taking into account other factors of memory cost such as model and optimizer state, hardware memory, and distributed partitioning, BPT allows training 2-4x larger context length. We plan to include the analysis into the revision.

---

> > ### Comment · Reviewer_t1yD · 2023-08-18
> > **Response to Rebuttal**
> >
> > Thanks for the analysis on memory cost. This addressed my question. I have upgraded my score.

---

> > > ### Author Response · Authors · 2023-08-18
> > > **Thank you for your response**
> > >
> > > We would like to thank the reviewer for raising their score and engaging during the rebuttal period. We thank the reviewer for detailed and positive assessment of our work.

---

### Official Review · Reviewer_DQeN · 2023-07-23

**Soundness:** 3 good
**Presentation:** 2 fair
**Contribution:** 3 good
**Rating:** 7
**Confidence:** 3

**Summary:**

The paper presents a novel variant of the Transformer model, the Blockwise Feedforward Transformer (BFT), which is designed to address the memory inefficiency issue in standard Transformer models. The authors propose a blockwise computation method that significantly reduces the memory footprint without compromising the model's performance.

The key innovation of the BFT is the introduction of a blockwise computation method. Instead of computing the feedforward network on the full sequence, the BFT computes it on intermediate blocks, resulting in substantial memory savings. The computation for a query block is given by: Outputi = FFN(Attention(Qi, K, V ) + Qi) + Attention(Qi, K, V ) + Qi.



**Strengths:**

1. The Blockwise Feedforward Transformer (BFT) presents a novel approach to address the memory inefficiency issue in standard Transformer models. This is a significant contribution as memory efficiency is a critical factor for training long context language models
2. The authors conducted extensive experiments on both reinforcement learning and language modeling tasks, providing a comprehensive evaluation of the model's performance across different domains.


**Weaknesses:**

1. The paper lacks a comprehensive analysis of the model's performance, particularly in terms of perplexity or its impact on downstream tasks. This omission makes it challenging to assess the effectiveness of the BFT in maintaining performance when compared to other models.



**Questions:**

1. Implementation Details: More information on the implementation details of the BFT would be appreciated. Are there any specific hardware/software requirements/dependencies for implementing the BFT?


**Limitations:**

The authors argue that "The application of long context models to NLP training remains uncertain due to the scarcity of large datasets encompassing extensive context information." However, resources such as the Books Corpus, Github, and Arxiv offer a wealth of long documents. It would be beneficial if the authors could experiment with training their proposed models using these resources and conduct a more thorough evaluation of the long context capabilities of Language Models.

---

> ### Author Rebuttal · Authors · 2023-08-09
>
> We thank the reviewer for their feedback and for a positive assessment of the work. We found the reviewer’s questions and suggestions insightful, and we list our plans to incorporate them below. Please let us know if our answers address your questions.
>
>
> > Q: The paper lacks a comprehensive analysis of the model's performance, particularly in terms of perplexity or its impact on downstream tasks. This omission makes it challenging to assess the effectiveness of the BFT in maintaining performance when compared to other models.
>
> A: For reviewer requested language model tasks, we have attached results in the table. The results are validation loss of GPT-XL (1B) on OpenWebText dataset. For shorter context length, BPT shows the validation loss as vanilla transformers and flashattention baselines, and BPT surpasses these baselines in terms of achieving a lower validation loss with extended context length.
>
> | Model              | Context Len | Val Loss | Throughput | SpeedUp |
> | ------------------ | ----------- | -------- | ---------- | ------- |
> | Vanila Attention   | 2048        | 2.46     | 3827       | 1x      |
> | FlashAttention     | 2048        | 2.46     | 4371       | 1.14x   |
> | Blockwise Parallel | 2048        | 2.46     | 3985       | 1.04x   |
> | Vanila Attention   | 4096        | 2.44     | 2340       | 1x      |
> | FlashAttention     | 4096        | 2.44     | 2567       | 1.10x   |
> | Blockwise Parallel | 4096        | 2.44     | 2687       | 1.15x   |
> | Vanila Attention   | 8192        | oom      | 2455       | 1x      |
> | FlashAttention     | 8192        | 2.43     | 2781       | 1.13x   |
> | Blockwise Parallel | 8192        | 2.43     | 2875       | 1.17x   |
> | Vanila Attention   | 16384       | oom      | 1701       | 1x      |
> | FlashAttention     | 16384       | oom      | 1889       | 1.11x   |
> | Blockwise Parallel | 16384       | 2.41     | 2045       | 1.20x   |
>
>
> > Q: Implementation Details: More information on the implementation details of the BFT would be appreciated. Are there any specific hardware/software requirements/dependencies for implementing the BFT?
>
> A: We plan to make all BPT code available upon acceptance. As for the hardware prerequisites, our experiments were conducted on various setups, including single-device and multi-host configurations, for both GPUs (A100) and TPUs (v3 and v4), therefore BPT should work on most GPUs and TPUs.
>
>
> > Q: The authors argue that "The application of long context models to NLP training remains uncertain due to the scarcity of large datasets encompassing extensive context information." However, resources such as the Books Corpus, Github, and Arxiv offer a wealth of long documents. It would be beneficial if the authors could experiment with training their proposed models using these resources and conduct a more thorough evaluation of the long context capabilities of Language Models.
>
>
> A: Undoubtedly, the potential to leverage extensive long context data is substantial. However, the current landscape presents specific challenges that warrant careful consideration. While resources like the Books Corpus, Github, and Arxiv offer a plethora of lengthy documents, effectively harnessing these resources demands addressing some key points.
>
> For instance, take the case of code repositories like Jax and Pytorch, which encompass a remarkable volume of approximately **120 million and 150 million tokens** respectively, excluding dependencies. These repositories harbor a wealth of contextual information. However, public Github datasets often contain only individual files rather than complete repositories, leading to an average document length significantly shorter than what models like BPT are capable of handling. Consider the 'starcoderdata' dataset, where the average document length is approximately 1500 tokens.
>
> Since the authors' primary focus is on addressing architecture bottlenecks, dealing with dataset challenges remains an ancillary pursuit.

---

### Official Review · Reviewer_wLgp · 2023-07-27

**Soundness:** 3 good
**Presentation:** 3 good
**Contribution:** 2 fair
**Rating:** 7
**Confidence:** 4

**Summary:**

"Blockwise Parallel Transformer for Large Models" presents a novel approach to handle the memory demands of Transformer models, mainly when dealing with long sequences or tasks involving multiple sequences or long-term dependencies. The authors propose a method called Blockwise Parallel Transformer (BPT) that leverages blockwise computation of self-attention and feedforward network fusion to minimize memory costs.

The critical contributions of the paper are:

1. The authors propose a blockwise computation of self-attention and feedforward approach that enables 16 to 64 times longer and 2 to 4 times longer context lengths than vanilla Transformer and previous memory-efficient Transformers, respectively.
2. They demonstrate the effectiveness of their approach through empirical experiments. BPT can reduce the memory requirements of Transformers, enabling training at least 8 to 64 times longer sequence than vanilla attention-based GPT models and at least 2 to 4 times longer sequence than prior state-of-the-art FlashAttention and Memory Efficient Attention.
3. The authors also show that BPT significantly improves performance and achieves better results on challenging reinforcement learning benchmarks by conditioning on multiple trajectories.

The authors argue that their approach has the potential to enable the training and evaluation of more complex models that require longer input sequences, which could lead to further breakthroughs in AI research.

The authors also explain the memory bottleneck of Transformer models and how their proposed method addresses this issue. They explain how self-attention can be computed blockwise without materializing the softmax attention matrix, resulting in significant memory savings. They also show how this blockwise computation can be applied to the feedforward network, resulting in further memory savings. The paper includes a detailed algorithm of the BPT method and key parts of its implementation in Jax which is nice to see.

**Strengths:**

**Originality:** BPT enables processing longer input sequences while maintaining or improving performance. This is achieved by computing both the feedforward network and self-attention blockwise, significantly reducing memory requirements (Page 9). This approach is original as it combines the techniques of blockwise computation of self-attention and feedforward networks, which is a distinct approach from previous works.

**Quality:** The paper demonstrates the effectiveness of the BPT approach from the perspective of speed/throughput (but not quality/capability of the model) through empirical experiments. It shows that the BPT approach can train on 8 to 64 times longer sequences than vanilla attention-based GPT models and 2 to 4 times longer sequences than prior state-of-the-art models like FlashAttention and Memory Efficient Attention.

**Clarity:** The paper is well-structured and clear in its presentation. It provides a comprehensive background on Transformers and their challenges in handling long sequences. The paper also clearly explains the proposed BPT approach and its advantages, such as its ability to leverage hardware with significantly faster SRAM speed compared to HBM speed. The evaluation of the method is also clearly presented with comparisons to other attention mechanisms.

**Significance:** The paper's significance lies in its potential to enable the training and evaluation of more complex models that require longer input sequences.

**Weaknesses:**

The use of RL as a benchmark for testing the quality of BPT, is a weird choice to say the least, especially given how noisy we know RL is. Language modeling is the primary use case for transformers and by extension BPT. Therefore, evaluating BPT's performance (perplexity) on language modeling tasks would be more appropriate.

I initially misunderstood that this is not an approximation but an exact implementation of attn+ffn (similar to flash attention). Therefore it's not imperative that there's LLM benchmarks but regardless would be nice to see that this implementation gives exact same output as standard attention in all precisions int8-fp32.

**Questions:**

- How does it work with parallelism schemes that chunk on sequence or FFN?
- Does this type of parallelism work with lower precision since it seems like there's more steps for accumulation and hence likely more noisy.

**Limitations:**

**Dependence on Hardware:** The paper mentions that the BPT can leverage hardware with significantly faster SRAM speed compared to HBM speed (Page 5). This implies that the performance of BPT might be dependent on the specific hardware configuration, and may not perform as well on hardware with slower SRAM speed.

**Sequence/Tensor Level Parallelism:** How does it work with parallelism schemes that chunk on sequence or FFN.

---

> ### Author Rebuttal · Authors · 2023-08-09
>
> We thank the reviewer for their feedback and for a positive assessment of the work. We found the reviewer’s questions and suggestions insightful, and we list our plans to incorporate them below. Please let us know if our answers address your questions.
>
>
> > Q: The use of RL as a benchmark for testing the quality of BPT, is a weird choice to say the least, especially given how noisy we know RL is. Language modeling is the primary use case for transformers and by extension BPT. Therefore, evaluating BPT's performance (perplexity) on language modeling tasks would be more appropriate.
>
>
> A: The primary reason for using this benchmark is training transformers across trajectories provides an easy way to set up a meaningful problem setting that requires challenging long context length.  While RL tasks in general exhibit significant noise, it's important to note that the ExoRL benchmark employed within this paper distinguishes itself by possessing a relatively lower level of noise. This attribute is attributed to the benchmark's utilization of *millions of diverse trajectories* for training, alongside the paper’s focus on training an agentic transformer using MLE loss.
>
> For reviewer requested language model tasks, we have attached results in the table. The results are validation loss of GPT-XL (1B) on OpenWebText dataset. For shorter context length, BPT shows the validation loss as vanilla transformers and flashattention baselines, and BPT surpasses these baselines in terms of achieving a lower validation loss with extended context length.
>
> | Model              | Context Len | Val Loss | Throughput | SpeedUp |
> | ------------------ | ----------- | -------- | ---------- | ------- |
> | Vanila Attention   | 2048        | 2.46     | 3827       | 1x      |
> | FlashAttention     | 2048        | 2.46     | 4371       | 1.14x   |
> | Blockwise Parallel | 2048        | 2.46     | 3985       | 1.04x   |
> | Vanila Attention   | 4096        | 2.44     | 2340       | 1x      |
> | FlashAttention     | 4096        | 2.44     | 2567       | 1.10x   |
> | Blockwise Parallel | 4096        | 2.44     | 2687       | 1.15x   |
> | Vanila Attention   | 8192        | oom      | 2455       | 1x      |
> | FlashAttention     | 8192        | 2.43     | 2781       | 1.13x   |
> | Blockwise Parallel | 8192        | 2.43     | 2875       | 1.17x   |
> | Vanila Attention   | 16384       | oom      | 1701       | 1x      |
> | FlashAttention     | 16384       | oom      | 1889       | 1.11x   |
> | Blockwise Parallel | 16384       | 2.41     | 2045       | 1.20x   |
>
> > Q: Does BPT work with lower precision since it seems like there's more steps for accumulation and hence likely more noisy?
>
> A: BPT works with lower precision, this is because in the attention part, BPT employs a numerically stable softmax, similar to the traditional standard transformer. It achieves this by maintaining a record of the largest value within each block during the attention phase and subsequently re-normalizing it using the overall maximum value; and in the FFN part, the blockwise computation for MLP does not incur potential numerical errors because it’s position wise. Our experimentation involved a comparison between BPT and a conventional standard transformer using various precisions, ranging from int8 to fp32. This evaluation was done with the provided code snippet.
>
> ```
> for dtype in [jnp.float16, jnp.float32, jnp.int8, jnp.bfloat16]:
>         for seed in range(10):
>             batch, seq_len, n_heads, head_dim = 4, 1024, 48, 64
>             key = jax.random.PRNGKey(seed)
>             if dtype == jnp.int8:
>                 inputs = jnp.random.randint(key, (batch, seq_len, n_heads * head_dim), -256, 256, dtype=dtype)
>             else:
>                 inputs = jnp.random.normal(key, (batch, seq_len, n_heads * head_dim), dtype=dtype)
>             variables = bpt_block.init(key, inputs)
>             o_bpt = bpt_block.apply(variables, inputs).block_until_ready()
>             variables = vanilla_block.init(key, inputs)
>             o_vanilla = vanilla_block.apply(variables, inputs).block_until_ready()
>             np.testing.assert_allclose(o_bpt, o_vanilla, rtol=1e-5, atol=1e-5)
> ```
>
> > Q: Does BPT work with sequence / tensor parallelism schemes?
>
> A: BPT is compatible with sequence parallelism and tensor parallelism. The sequence parallelism distributes sequence dimension across devices, in contrast, BPT implements blockwise computation on sequences for each device. This creates an orthogonal relationship between our method and sequence parallelism, allowing for straightforward combination. BPT is compatible with both tensor and data parallelism since BPT does not change the computation along tensor and batch dimensions.
>
>
> > Dependence on Hardware: The paper mentions that the BPT can leverage hardware with significantly faster SRAM speed compared to HBM speed (Page 5). This implies that the performance of BPT might be dependent on the specific hardware configuration, and may not perform as well on hardware with slower SRAM speed.
>
> A: On modern accelerators, the speed of SRAM vastly surpasses that of HBM. Consequently, BPT offers the dual benefits of accelerated speed and efficient memory utilization. However, it's important to note that in cases where the hardware features slower SRAM speed, BPT still remains advantageous in terms of memory optimization, even if the degree of speed enhancement is not as pronounced.

---

### Decision · Program_Chairs · 2023-09-21

**Decision:**

Accept (spotlight)

**Comment:**

The paper introduces an efficient blockwise computation scheme for transformers. Instead of fully materializing the NxN attention matrix, the computation for queries is fused with subsequent feedforward layers, which significantly improves memory efficiency. Importantly, the method outputs the exact same result as standard transformers. All the reviewers agreed that this is a straightforward but well motivated contribution, and it seems very practically useful, so this is a clear accept.